# Use of hyperbaric oxygen therapy for preventing delayed neurological sequelae in patients with carbon monoxide poisoning: A multicenter, prospective, observational study in Japan

**Motoki Fujita** [1]*, **Masaki Todani**[2], **Kotaro Kaneda**[2], **Shinya Suzuki**[3], **Shinjiro Wakai**[4], **Shota Kikuta**[5], **Satomi Sasaki**[6], **Noriyuki Hattori**[7], **Kazuyoshi Yagishita**[8], **Koji Kuwata**[9], **Ryosuke Tsuruta**[1,2], **on behalf of the COP-J Study Investigators**[¶]

1 Acute and General Medicine, Yamaguchi University Graduate School of Medicine, Ube, Japan, 2 Advanced Medical Emergency and Critical Care Center, Yamaguchi University Hospital, Ube, Japan, 3 Department of Emergency Medicine, Kameda Medical Center, Kamogawa, Japan, 4 Department of Emergency and Critical Care Medicine, Tokai University School of Medicine, Isehara, Japan, 5 Department of Emergency and Critical Care Medicine, Hyogo Emergency Medical Center, Kobe, Japan, 6 Advanced Medical Emergency Department and Critical Care Center, Japanese Red Cross Maebashi Hospital, Maebashi, Japan, 7 Department of Emergency and Critical Care Medicine, Chiba University Graduate School of Medicine, Chiba, Japan, 8 Hyperbaric Medical Center, Tokyo Medical and Dental University, Tokyo, Japan, 9 Division of Medicine, Japan Self Defense Forces Hospital Yokosuka, Yokosuka, Japan

¶ Details of the COP-J Study Investigators are indicated in the Acknowledgments.
* motoki-ygc@umin.ac.jp

**Data Availability Statement:** All relevant data are within the paper.

## Abstract

### Background

The purpose of this study was to clarify the practical clinical treatment for acute carbon monoxide (CO) poisoning in Japan and to investigate the efficacy of hyperbaric oxygen ($HBO_2$) therapy in preventing delayed neurological sequelae (DNS) in the acute phase of CO poisoning.

### Methods

We conducted a multicenter, prospective, observational study of acute CO poisoning in Japan. Patients with acute CO poisoning were enrolled and their treatment details were recorded. The primary endpoint was the onset of DNS within 2 months of CO exposure. Factors associated with DNS were assessed with logistic regression analysis.

### Results

A total of 311 patients from 57 institutions were registered and 255 were analyzed: 171 received $HBO_2$ therapy ($HBO_2$ group) and 84 did not (normobaric oxygen [$NBO_2$] group). $HBO_2$ therapy was performed zero, once, twice, or three times within the first 24 h in 1.8%, 55.9%, 30.9%, and 11.3% of the $HBO_2$ group, respectively. The treatment pressure in the first $HBO_2$ session was 2.8 ATA (47.9% of the $HBO_2$ group), 2.0 ATA (41.8%), 2.5 ATA (7.9%), or another pressure (2.4%). The incidence of DNS was 13/171 (7.6%) in the $HBO_2$ group and 3/84 (3.6%) in the $NBO_2$ group ($P = 0.212$). The number of $HBO_2$ sessions in the

**Funding:** The authors received no specific funding for this work.

**Competing interests:** The authors have declared that no competing interests exist.

first 24 h was one of the factors associated with the incidence of DNS (odds ratio, 2.082; 95% confidence interval, 1.101–3.937; $P$ = 0.024).

## Conclusions

The practical clinical treatment for acute CO poisoning, including $HBO_2$ therapy, varied among the institutions participating in Japan. $HBO_2$ therapy with inconsistent protocols showed no advantage over $NBO_2$ therapy in preventing DNS. Multiple $HBO_2$ sessions was associated with the incidence of DNS.

## Introduction

Hyperbaric oxygen ($HBO_2$) therapy is thought to be essential for preventing neurological sequelae in patients with carbon monoxide (CO) poisoning, based on the results of a randomized controlled trial (RCT) reported by Weaver et al. [1]. However, the results of RCTs, including subsequent reports, have been conflicting [2–6], and the effects of $HBO_2$ therapy for patients with CO poisoning remains contentious. A previous meta-analysis did not find beneficial effects of $HBO_2$ therapy or the reduction of adverse neurological outcomes by $HBO_2$ therapy for CO poisoning [7]. Therefore, it is unclear whether $HBO_2$ therapy in the acute phase of CO poisoning prevents neurological sequelae.

Our previous survey, performed by questionnaire, showed that the clinical practice of $HBO_2$ therapy for CO poisoning varied in both its indications and the practice regimens used in Japan [8]. This situation is not specific to Japan and has also been reported in the USA and Europe [9, 10]. These findings suggest that there is no clear clinical consensus about $HBO_2$ therapy for acute CO poisoning. Therefore, we conducted a multicenter, prospective, observational study of acute CO poisoning to clarify the practical clinical treatment for acute CO poisoning in Japan and to investigate the efficacy of $HBO_2$ therapy in preventing DNS in the acute phase of CO poisoning.

## Methods

### Design and setting

We conducted a multicenter, prospective, observational study of acute CO poisoning in Japan called the COP-J Study to clarify the efficacy of $HBO_2$ therapy in the acute phase of CO poisoning. A cohort of patients with acute CO poisoning from 54 institutions was enrolled in the COP-J Study, which recorded the patients' data after approval was given by the Ethics Committee of each institution. The COP-J Study was approved by the Japanese Society of Intensive Care Medicine (No. 0011). The therapeutic policies of the majority of these institutions have already been reported [8] and 19 (35.2%) of the 54 institutions involved in this study did not administer $HBO_2$ therapy and performed only normobaric oxygen ($NBO_2$) therapy. The 35 enrolled institutions that had an $HBO_2$ chamber administered $HBO_2$ therapy according to their institutional policies [8]. At the start of the study, there were 568 institutions in Japan that had an $HBO_2$ chamber, of which 115 had a board-certified fellow of the Japanese Society of Hyperbaric and Undersea Medicine.

### Data collection and analysis

Patients diagnosed with acute CO poisoning based on any symptoms after CO exposure or on a carboxyhemoglobin (COHb) level exceeding 10%, between October 2015 and September

2018, were enrolled in the study. The medical records of the patients, including the circumstances of CO exposure, prehospital information, physical and laboratory findings upon arrival, and details of treatments, including $HBO_2$ therapy, were recorded by the University Hospital Medical Information Network–Internet Data and Information Center for Medical Research (UMIN–INDICE) web system. The primary endpoint was the onset of delayed neurological sequelae (DNS) within 2 months of CO exposure. DNS was defined as cognitive dysfunction that affected daily life after an improvement in disturbed consciousness. DNS was checked at outpatient consultations or by telephone if the patient did not visit the hospital. In the telephone consultation for DNS, the physician addressed the following questions to the patients or their family: "Is there any hindrance to daily life?"; "Do you have memory problems?"; "Is there any change in your personality?"; "Are there more things you cannot do compared with before?", and so on. If there was any doubt about the presence of DNS, the physician instructed the patient to visit the hospital. DNS was finally diagnosed by a physician based on all the findings at the time of diagnosis, including results of a cognitive function test, such as the mini-mental state examination, the Wechsler adult intelligence scale, Hasegawa's dementia scale-revised [11], the trail-making test, or the story recall test. In addition, the physicians were not blinded as to the treatment of acute CO poisoning. The secondary endpoint was the improvement in prolonged consciousness disturbance (PCD), which was defined as prolonged consciousness disturbance after 24 h from admission. The improvement in PCD was checked by a physician at discharge or at 2 months after CO exposure. Before the analysis, we excluded patients with cardiopulmonary arrest upon arrival, or in-hospital death, or who were lost to follow-up. In the analysis, we compared the incidence of DNS and improvement in PCD between patients who received either $HBO_2$ or $NBO_2$ therapy during the acute phase. The factors associated with DNS and unimproved PCD were also identified.

## Statistical analyses

Variables are shown as means ± standard deviations or numbers (percentages). Missing values were excluded from all analyses. Univariate analyses were performed with a $t$ test for continuous variables and a $\chi^2$ test for categorical variables. Univariate regression and multivariable logistic regression with the stepwise variable selection method were performed to identify factors associated with DNS and unimproved PCD, and the results are presented as odds ratios (ORs) and 95% confidence intervals (CIs). The factors associated with DNS and unimproved PCD in previous reports [12–17] were included as variables in the multivariable logistic regression models. Values of $P < 0.05$ were considered to indicate statistical significance. All analyses were performed with IBM SPSS Statistics for Windows version 22 (IBM SPSS Inc., Chicago, IL).

## Results

### Patients' characteristics

A total of 311 patients from 54 institutions were registered and 255 were included in the analysis (Fig 1). Of the patients included, 171 received $HBO_2$ therapy ($HBO_2$ group) and 84 did not ($NBO_2$ group). Patients excluded from the analyses included 12 with cardiopulmonary arrest on arrival (CPAOA), three who died in hospital, and 41 who were lost to follow-up.

The patients' characteristics and the physiological and laboratory findings on arrival are shown in Table 1. The mean age was 54 ± 22 years in the $NBO_2$ group and 49 ± 19 years in the $HBO_2$ group ($P = 0.063$). Almost 60% of the patients were male and half the patients had a history of smoking. The sex ratios and smoking histories did not differ significantly between the $NBO_2$ and $HBO_2$ groups. The total rate of patients who had attempted suicide was 29.8% and

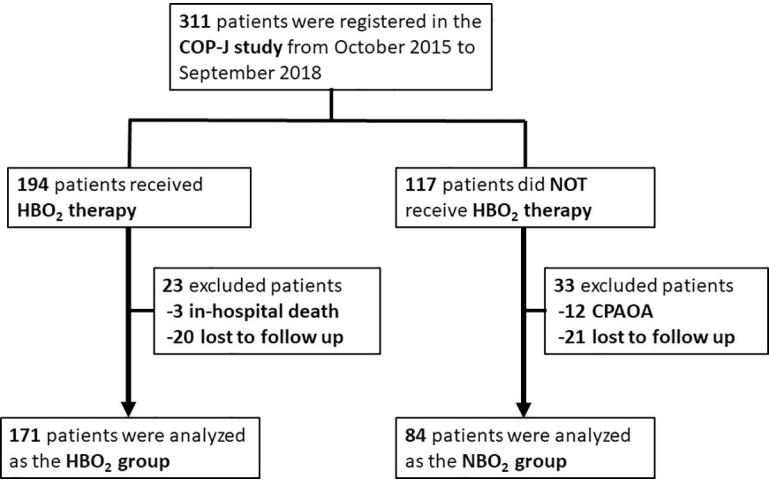

**Fig 1. Flowchart of patient selection.** HBO$_2$, hyperbaric oxygen; NBO$_2$, normobaric oxygen; CPAOA, cardiopulmonary arrest on arrival.

the difference between the NBO$_2$ and HBO$_2$ groups was not significant (25.0% vs 32.2%, respectively; $P = 0.240$). In more than half the patients in both groups, CO poisoning was caused by burning charcoal. In the NBO$_2$ group, the number of cases caused by fires was greater than in the HBO$_2$ group, whereas the number of cases caused by car exhausts was lower. The environmental circumstances of CO exposure was the same in both groups. Almost all the patients arrived at hospital by ambulance and the incidence of loss of consciousness was the same in the NBO$_2$ and HBO$_2$ groups (42.3% vs 48.0%, respectively, $P = 0.413$). Oxygen was administered by the emergency medical service slightly less frequently in the NBO$_2$ group than in the HBO$_2$ group (84.2% vs 92.4%, respectively; $P = 0.064$). The estimated time of exposure to CO was 181 ± 376 min in the NBO$_2$ group and 202 ± 256 min in the HBO$_2$ group, and the difference was not significant ($P = 0.605$). The time from CO exposure to hospitalization was the same between the NBO$_2$ and HBO$_2$ groups (240 ± 382 and 279 ± 350 min, respectively; $P = 0.420$). In the NBO$_2$ group, 47 (56.0%) patients were transferred to an institution that offered only NBO$_2$ therapy by EMS.

In the arterial blood gas analyses, PaO$_2$ was significantly lower in the NBO$_2$ group than in the HBO$_2$ group (198 ± 103 vs 270 ± 122 Torr, respectively; $P < 0.001$) and lactic acidosis was significantly more severe in the NBO$_2$ group than in the HBO$_2$ group. There was no significant difference in COHb levels between the NBO$_2$ and HBO$_2$ groups (19.3 ± 10.2% and 18.7 ± 11.4%, respectively; $P = 0.682$). Furthermore, in the NBO$_2$ group, the COHb levels were 20.9 ± 10.0% in patients who were transferred to institutions that only offered NBO$_2$ therapy and 17.2 ± 10.1% in patients transferred to institutions that also provided HBO$_2$ therapy ($P = 0.101$).

## Treatment regimens including HBO$_2$ therapy and NBO$_2$ therapy

The number of HBO$_2$ sessions during the first 24 h and the first week in the HBO$_2$ group are shown in Fig 2. HBO$_2$ therapy was performed zero, one, two, or three times within the first 24 h in 1.8%, 55.9%, 30.9%, and 11.3% of the HBO$_2$ group, respectively. In the HBO$_2$ group, 30 patients (17.9% of the group) received HBO$_2$ therapy only once during the first week after admission; 49 patients (29.2% of the HBO$_2$ group) received HBO$_2$ therapy three times in the first week; and the maximum number of treatments during the first week was 15. The average time from arrival to the first HBO$_2$ session was 158 ± 147 min among the patients who were administered HBO$_2$ therapy on the first day.

**Table 1. Patient characteristics and physiological and laboratory findings upon arrival.**

| | NBO2 (n = 84) | HBO2 (n = 171) | P-value |
|---|---|---|---|
| Age | 54 ±22 | 49 ± 19 | 0.063 |
| Sex (male, %) | 51 (60.7%) | 107 (62.6%) | 0.774 |
| Smoking | 37 (48.7%) | 67 (47.5%) | 0.870 |
| Type | | | 0.240 |
| Accidental | 63 (75.0%) | 116 (67.8%) | |
| Intentional | 21 (25.0%) | 55 (32.2%) | |
| Cause | | | <0.001 |
| Charcoal | 43 (51.2%) | 87 (50.9%) | 0.963 |
| Fire | 26 (31.0%) | 12 (7.0%) | <0.001 |
| Car exhaust | 5 (6.0%) | 34 (19.9%) | 0.002 |
| Other | 10 (11.8%) | 38 (22.2%) | 0.041 |
| Environment | | | 0.097 |
| Indoor | 68 (81.0%) | 122 (71.3%) | |
| Outdoor | 5 (6.0%) | 7 (4.1%) | |
| In a car | 11 (13.0%) | 42 (24.6%) | |
| Arrived by ambulance | 78 (92.9%) | 150 (87.7%) | 0.210 |
| Loss of consciousness | 33 (42.3%) | 72 (48.0%) | 0.413 |
| Oxygen administration by EMS | 64 (84.2%) | 122 (92.4%) | 0.064 |
| SpCO (%) at scene | 26.4 ± 20.9, (n = 8) | 30.1 ± 15.7, (n = 40) | 0.562 |
| Exposure time (min) | 181 ± 376 | 202 ± 256 | 0.605 |
| Time from exposure to hospital (min) | 240 ± 382 | 279 ± 350 | 0.420 |
| No. patients transferred to the institution performing only NBO$_2$ | 47 (56.0%) | | |
| Glasgow Coma Scale on arrival | 13 ± 4 | 13 ± 3 | 0.445 |
| Systolic blood pressure (mmHg) | 138 ± 30 | 133 ± 23 | 0.223 |
| Diastolic blood pressure (mmHg) | 79 ± 20 | 77 ± 17 | 0.497 |
| Heart rate (/min) | 100 ± 26 | 87 ± 19 | <0.001 |
| Respiratory rate (/min) | 21 ± 7 | 20 ± 5 | 0.071 |
| Body temperature (°C) | 36.5 ± 0.9 | 36.7 ± 0.7 | 0.046 |
| Blood gas analysis (BGA) | | | |
| Time from arrival to BGA (min) | 11.4 ± 12.2 | 14.0 ± 17.4 | 0.257 |
| pH | 7.374 ± 0.102 | 7.409 ± 0.073 | 0.008 |
| PaO$_2$ (Torr) | 198 ± 103 | 270 ± 122 | <0.001 |
| PaCO$_2$ (Torr) | 37.9 ± 19.4 | 36.2 ± 6.9 | 0.342 |
| HCO$_3^-$ (mmol/L) | 21.0 ± 4.6 | 22.8 ± 4.1 | 0.003 |
| Base excess (mmol/L) | -3.2 ± 5.9 | -1.3 ± 4.6 | 0.016 |
| Lactate (mmol/L) | 4.7 ± 3.9 | 3.5 ± 4.3 | 0.045 |
| SaO$_2$ (%) | 97.2 ± 5.8 | 97.2 ± 6.5 | 0.979 |
| COHb (%) | 19.3 ± 10.2 | 18.7 ± 11.4 | 0.682 |
| MetHb (%) | 0.8 ± 0.6 | 1.2 ± 1.7 | 0.071 |
| Hematocrit | 41.9 ± 5.8 | 41.9 ± 5.6 | 0.980 |
| White blood cell (/μL) | 10483 ± 5395 | 10195 ± 5019 | 0.685 |
| C-reactive protein (mg/dL) | 0.9 ± 3.0 | 0.7 ± 2.5 | 0.637 |
| Creatine kinase (IU/L) | 361 ± 1000 | 567 ± 2295 | 0.446 |
| Creatine kinase MB fraction (IU/L) | 14.0 ± 13.3, (n = 40) | 13.9 ± 43.0, (n = 89) | 0.993 |
| Above normal range | 11 (27.5%), (n = 40) | 12 (13.5%), (n = 89) | 0.061 |
| Troponin T, positive | 7 (29.2%), (n = 24) | 18 (22.8%), (n = 79) | 0.529 |
| ECG abnormality | 18 (22.8%), (n = 79) | 33 (20.9%), (n = 158) | 0.737 |

*(Continued)*

**Table 1.** (Continued)

|  | NBO2 (n = 84) | HBO2 (n = 171) | *P-value* |
|---|---|---|---|
| ST-T change | 9 (50.0%) | 10 (30.3%) | |
| AF rhythm | 3 (16.7%) | 3 (9.1%) | |
| Other | 6 (33.3%) | 20 (60.6%) | |
| Abnormal findings on CT | 6 (10.7%), (*n* = 58) | 17 (15.9%), (*n* = 107) | 0.368 |
| Lesion(s) on basal ganglia | 1 (16.7%) | 11 (64.7%) | |
| Acute cerebral infarction | 1 (16.7%) | 0 (0%) | |
| Chest lesion | 3 (50.0%) | 1 (5.9%) | |
| Other | 1 (16.7%) | 5 (29.4%) | |
| Abnormal findings on head MRI | 7 (22.6%), (*n* = 32) | 24 (23.5%), (*n* = 106) | 0.913 |
| Lesion(s) on basal ganglia or white matter | 4 (57.1%) | 19 (79.2%) | |
| Other | 3 (42.9%) | 5 (20.8%) | |

NBO$_2$, normobaric oxygen; HBO$_2$, hyperbaric oxygen; EMS, emergency medical service; SpCO, carbon monoxide hemoglobin saturation; COHb, carboxyl hemoglobin; MetHb, methemoglobin; ECG, electrocardiogram; AF, atrial fibrillation; CT, computed tomography; MRI, magnetic resonance imaging.

The treatment pressures in each HBO$_2$ session during the first 24 h are shown in Table 2. The treatment pressure in the first HBO$_2$ session was 2.8 atmospheres absolute (ATA) (47.9% of the HBO$_2$ group), 2.0 ATA (41.8%), 2.5 ATA (7.9%), or another pressure (2.4%). A treatment pressure of 2.0 ATA was used in the majority of patients in both the second and third HBO$_2$ sessions. In addition, HBO$_2$ therapy were not administered during the first 24 h in 2 patients of the HBO$_2$ group and the details of HBO$_2$ therapy were unknown in 4 patents.

The number of patients treated with mechanical ventilation was significantly more in the NBO$_2$ group than in the HBO$_2$ group (25.0% vs 4.7%, respectively, P < 0.001; Table 3). The period of oxygen administration during the hospital stay was 344 ± 2128 h and 70 ± 190 h in the NBO$_2$ and HBO$_2$ groups, respectively, which did not differ significantly (*P* = 0.266; Table 3). ICU days was also significantly longer in the NBO$_2$ group than in the HBO$_2$ groups (4.1 ± 11.0 and 1.3 ± 2.4 days, respectively; *P* = 0.025; Table 3), but hospital days did not differ between the groups (*P* = 0.294; Table 3).

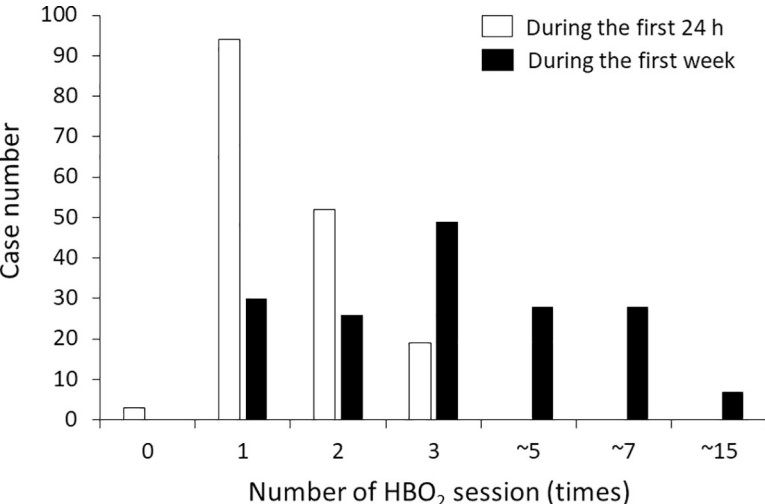

**Fig 2. Number of HBO$_2$ session during the first 24 h and the first week in the HBO$_2$ group.**

Table 2. Treatment pressure in each HBO$_2$ session during the first 24 h.

| Treatment pressure | First ($n$ = 165) | Second ($n$ = 71) | Third ($n$ = 19) |
|---|---|---|---|
| 1.5 ATA | 1 (0.6%) | | |
| 2.0 ATA | 69 (41.8%) | 38 (53.5%) | 12 (63.2%) |
| 2.1 ATA | | | 2 (10.5%) |
| 2.4 ATA | 2 (1.2%) | 11 (15.5%) | 4 (21.1%) |
| 2.5 ATA | 13 (7.9%) | 9 (12.7%) | |
| 2.7 ATA | 1 (0.6%) | | |
| 2.8 ATA | 79 (47.9%) | 13 (18.3%) | 1 (5.3%) |

ATA, atmospheres absolute.

## Incidence of DNS, improvement in PCD, and factors associated with DNS and unimproved PCD

The total incidence of DNS was 16/255 (6.3%) in this study, and did not differ between the NBO$_2$ group and the HBO$_2$ group (3.6% vs 7.6%, respectively; $P$ = 0.212, Table 3). The total incidences of PCD and unimproved PCD were 19/255 (7.5%) and 8/255 (3.1%), respectively. Neither of these measures differed between the NBO$_2$ group and the HBO$_2$ group (PCD: 6.0% vs 8.2%, respectively, $P$ = 0.523; unimproved PCD: 2.4% vs 3.5%, respectively, $P$ = 0.627; Table 3).

Concerning the association between the number of HBO$_2$ sessions in the first 24 h and the incidence of DNS, a greater number of HBO$_2$ sessions in the first 24 h was associated with a greater incidence of DNS ($P$ = 0.020; Table 4). The incidence of unimproved PCD was not associated with the number of HBO$_2$ sessions in the first 24 h ($P$ = 0.735; Table 4).

The treatment pressures in the first HBO$_2$ session were 2.8 ATA ($n$ = 7), 2.5 ATA ($n$ = 1), and 2.0 ATA ($n$ = 4) in the DNS patients in the HBO$_2$ group, and 2.8 ATA ($n$ = 6) and 2.0 ATA ($n$ = 1) in the unimproved PCD patients in the HBO$_2$ group.

Among 35 patients with abnormal findings in CT or MRI, DNS was observed in 2 (22.2%) and 8 (30.8%) patients in the NBO$_2$ group ($n$ = 9) and the HBO$_2$ group ($n$ = 26), respectively. There was no significant difference in the incidence of DNS between the groups ($P$ = 0.625). Unimproved PCD was observed in 2 (22.2%) and 6 (23.1%) patients in the NBO$_2$ group and the HBO$_2$ group, respectively. There was no significant difference between the groups ($P$ = 0.958).

Table 3. Therapeutic periods and incidence of neurological sequelae.

| | NBO$_2$ ($n$ = 84) | HBO$_2$ ($n$ = 171) | $P$ value |
|---|---|---|---|
| MV | 21 (25.0%) | 8 (4.7%) | <0.001 |
| Period of MV (h) | 557 ± 3157 | 6 ± 31 | 0.127 |
| Period of oxygen administration during the hospital stay (h) | 344 ± 2128 | 70 ± 190 | 0.266 |
| ICU stay (days) | 4.1 ± 11.0 | 1.3 ± 2.4 | 0.025 |
| Hospital stay (days) | 15.2 ± 25.5 | 11.1 ± 30.2 | 0.294 |
| DNS | 3 (3.6%) | 13 (7.6%) | 0.212 |
| PCD | 5 (6.0%) | 14 (8.2%) | 0.523 |
| Unimproved PCD | 2 (2.4.%) | 6 (3.5%) | 0.627 |

NBO$_2$, normobaric oxygen; HBO$_2$, hyperbaric oxygen; MV, mechanical ventilation; ICU, intensive care unit; DNS, delayed neurological sequelae; PCD, prolonged consciousness disturbance.

**Table 4. Number of HBO$_2$ therapy sessions in the first 24 h and incidence of neurological sequelae.**

| No. of HBO$_2$ sessions in the first 24 h | n (%) | DNS | Unimproved PCD |
|---|---|---|---|
| 0 | 87 (34.5%) | 4 (4.6%) | 2 (2.3%) |
| 1 | 94 (37.3%) | 3 (3.2%) | 4 (4.3%) |
| 2 | 52 (20.6%) | 5 (9.6%) | 2 (3.8%) |
| 3 | 19 (7.5%) | 4 (21.1%) | 0 (0%) |
| | | P = 0.020 | P = 0.735 |

HBO$_2$, hyperbaric oxygen; DNS, delayed neurological sequelae; PCD, prolonged consciousness disturbance.

The following variables, previously reported to be associated with DNS and unimproved PCD [12–17], were included in the univariate and multivariable logistic regression models to identify factors associated with the incidence of DNS and unimproved PCD: age, sex, type of CO poisoning, cause, consciousness loss at the scene, estimated exposure time, time from exposure to hospital, Glasgow Coma Scale (GCS) score on arrival, COHb, lactate level, white blood cell count, and number of HBO$_2$ sessions and maximum therapeutic pressure in the first 24 h.

In the univariate regression analysis for the incidence of DNS, type of CO poisoning (intentional), cause (charcoal), consciousness loss at the scene, estimated exposure time, time from exposure to hospital, GCS score on arrival, white blood cell count, and number of HBO$_2$ sessions in the first 24 h were statistically significant (Table 5). The exposure time (OR, 1.003; 95% CI, 1.001–1.004; $P < 0.001$), GCS score (OR, 0.803; 95% CI, 0.695–0.927; $P = 0.003$), and the number of HBO$_2$ sessions in the first 24 h (OR, 2.082; 95% CI, 1.101–3.937; $P = 0.024$) were independently associated with the incidence of DNS in the multivariable logistic regression model (Table 5).

**Table 5. Factors associated with the incidence of delayed neurological sequelae (DNS).**

| | Univariate regression analysis | | | Multivariable logistic regression analysis | | |
|---|---|---|---|---|---|---|
| | OR | 95% CI | P value | OR | 95% CI | P value |
| Age (years) | 1.010 | 0.985–1.036 | 0.446 | | | |
| Sex, male | 1.911 | 0.598–6.102 | 0.274 | | | |
| Type, intentional | 0.170 | 0.057–0.507 | 0.002 | | | |
| Cause | | | | | | |
| Charcoal | 16.174 | 2.103–124.398 | 0.007 | | | |
| Fire | 0.364 | 0.047–2.840 | 0.335 | | | |
| Car exhaust | 0.000 | 0.000 | 0.993 | | | |
| Other | 0.000 | 0.000 | 0.997 | | | |
| Consciousness loss at the scene | 3.839 | 1.199–12.290 | 0.023 | | | |
| Estimated exposure time (min) | 1.004 | 1.002–1.005 | <0.001 | 1.003 | 1.001–1.004 | <0.001 |
| Time from exposure to hospital (min) | 1.002 | 1.001–1.003 | <0.001 | | | |
| Glasgow Coma Scale on arrival | 0.791 | 0.710–0.883 | <0.001 | 0.803 | 0.695–0.927 | 0.003 |
| COHb (%) | 1.020 | 0.974–1.068 | 0.406 | | | |
| Lactate (mmol/L) | 1.039 | 0.931–1.159 | 0.495 | | | |
| White blood cells (×10$^3$/μL) | 1.116 | 1.039–1.193 | 0.003 | | | |
| Number of HBO$_2$ sessions in the first 24 h | 1.891 | 1.120–3.192 | 0.017 | 2.082 | 1.101–3.937 | 0.024 |
| Maximum therapeutic pressure in first 24 h | 1.476 | 0.725–3.008 | 0.283 | | | |

OR, odds ratio; CI, confidence interval; COHb, carboxyl hemoglobin; HBO$_2$, hyperbaric oxygen.

**Table 6. Factors associated with the incidence of unimproved PCD.**

| | Univariate regression analysis | | | Multivariable logistic regression analysis | | |
|---|---|---|---|---|---|---|
| | OR | 95% CI | *P* value | OR | 95% CI | *P* value |
| Age (year) | 0.993 | 0.970–1.016 | 0.535 | | | |
| Sex, male | 0.661 | 0.258–1.689 | 0.387 | | | |
| Type, intentional | 0.914 | 0.334–2.501 | 0.860 | | | |
| Cause | | | | | | |
| Charcoal | 2.205 | 0.810–6.001 | 0.122 | | | |
| Fire | 1.119 | 0.309–4.054 | 0.864 | | | |
| Car exhaust | 0.281 | 0.036–2.166 | 0.223 | | | |
| Other | 0.484 | 0.108–2.170 | 0.864 | | | |
| Consciousness loss at the scene | 2.516 | 0.910–6.958 | 0.075 | | | |
| Estimated exposure time (min) | 1.002 | 1.001–1.004 | <0.001 | | | |
| Time from exposure to hospitalization (min) | 1.001 | 1.000–1.002 | 0.003 | 1.002 | 1.001–1.004 | 0.007 |
| Glasgow Coma Scale on arrival | 0.876 | 0.788–0.974 | 0.015 | | | |
| COHb (%) | 0.960 | 0.916–1.005 | 0.083 | | | |
| Lactate (mmol/L) | 1.051 | 0.949–1.163 | 0.338 | | | |
| White blood cells (×$10^3$/μL) | 1.001 | 0.999–1.097 | 0.987 | | | |
| Number of HBO$_2$ session in first 24 h | 1.353 | 0.834–2.196 | 0.221 | | | |
| Maximum therapeutic pressure in first 24 h | 1.954 | 0.969–3.940 | 0.061 | | | |

PCD, prolonged consciousness disturbance; OR, odds ratio; CI, confidence interval; COHb, carboxyl hemoglobin; HBO$_2$, hyperbaric oxygen.

In the univariate regression analysis for unimproved PCD, estimated exposure time, time from exposure to hospitalization, and GCS score on arrival were statistically significant (Table 6). The time from exposure to hospital (OR, 1.002; 95% CI, 1.001–1.004; *P* = 0.007) was independently associated with unimproved PCD in the multivariable logistic regression model (Table 6).

## Discussion

In this study, it has been shown that the clinical practice for acute CO poisoning varies in Japan, and that the application of and protocols for HBO$_2$ therapy are not consistent. HBO$_2$ therapy with inconsistent protocols showed no advantage over NBO$_2$ therapy in preventing DNS and unimproved PCD. Furthermore, a greater number of HBO$_2$ sessions in the first 24 h was associated with a higher incidence of DNS.

In clinical practice, the treatment for acute CO poisoning, including HBO$_2$ therapy, varied in the present study, as in our previous report [8]. In particular, the profiles of HBO$_2$ therapy, including the number of treatments given and the therapeutic pressures used, were not consistent. These results are similar to reports from Europe and the USA [9, 10], and may indicate that there is no global consensus on an effective regimen of HBO$_2$ therapy for CO poisoning. Further research, including RCTs, is required to establish consensus on these issues.

In the present study, the total incidence of DNS was only 6.3%, which is lower than that in other studies [1–6]. In our study, all of the patients with any symptoms after CO exposure or with a COHb level exceeding 10% were registered and analyzed, except for 12 CPAOA patients and three patients who died in hospital (Fig 1). The patients in this study might have had milder conditions than those in other studies because the entry criteria were less restrictive. Furthermore, in this study, DNS was only defined as cognitive dysfunction that affected daily life after an improvement in disturbed consciousness and did not include minor symptoms,

such as tinnitus or headache. Therefore, patients with mild symptoms or with symptoms other than cognitive dysfunction were not included. Furthermore, 40% of the patients without DNS were only diagnosed by telephone, so patients with mild symptoms might have been overlooked. These aspects of our study may have influenced the lower incidence of DNS.

Although the protocol for $HBO_2$ therapy varied, incidences of DNS and unimproved PCD did not differ between the patients treated with $NBO_2$ only and those treated with $HBO_2$, and the incidence of DNS tended to be lower in patients treated with $NBO_2$ only than in those treated with $HBO_2$ in this study (Table 3). Many RCTs have tried to clarify the efficacy of $HBO_2$ therapy in preventing DNS after CO poisoning [1–6], and half of them have shown no beneficial effects of $HBO_2$ therapy in this context [2, 3, 6]. In contrast, several reports have claimed that therapeutic pressure less than 2.5 ATA does not produce the beneficial effects of $HBO_2$ therapy [18, 19]. Thom et al. reported that the adherence of activated neutrophils, which is one of the mechanisms underlying the development of DNS after CO poisoning, was suppressed experimentally at 2.5 or 3.0 ATA, but not at 2.0 ATA [18]. The therapeutic pressures in the RCTs that demonstrated the beneficial effects of $HBO_2$ therapy exceeded 2.5 ATA [1, 4, 5, 20]. Birmingham and Hoffman claimed that inadequate pressure during $HBO_2$ therapy may only enhance oxygen toxicity, without the benefit offered by $HBO_2$ at higher pressures [19]. In the present study, only 60% of the patients in the $HBO_2$ group were administered the first session of $HBO_2$ therapy at pressures of more than 2.5 ATA (Table 2) and the same rate was observed in the DNS patients treated with $HBO_2$ therapy at pressures of more than 2.5 ATA. Therefore, in this study, insufficient treatment pressure might also have affected the number of patients with DNS.

Oxidative stress is a key mechanism in DNS [20–25]. $HBO_2$ reduced oxidative stress in an animal model of CO poisoning [26] and its beneficial effects included inhibition of leukocyte beta-2 integrins [18], reversal of CO-cytochrome c oxidase binding [27], and recovery of energy metabolism [28]. However, there have been reports that $HBO_2$ therapy itself induces oxidative stress [29–32]. Experimental data have shown that $HBO_2$ induces oxidative stress in healthy rat brains, measured as the lipid peroxidation products in brain cortex homogenates [29–31]. This $HBO_2$-induced oxidative stress is related to the $HBO_2$ pressure [29] or the exposure time [30]. It has also been reported that a single session of $HBO_2$ (2.4 kPa, 131 min) reduced plasma vitamin C and increased plasma lipid peroxides and urinary 8-oxo-deoxyguanosine excretion in healthy volunteers [32]. Although $HBO_2$ therapy has beneficial effects, it should be considered that there are concerns about adverse effects of $HBO_2$ therapy such as $HBO_2$-induced oxidative stress.

A greater number of $HBO_2$ sessions in the first 24 h was associated with a higher incidence of DNS (Tables 4 and 5). Two RCTs have reported that two $HBO_2$ sessions at 2.0 ATA were neither more beneficial nor more harmful than one session [2, 3], although multiple $HBO_2$ sessions at 2.5 to 2.8 ATA had beneficial effects on preventing DNS [1, 4, 5]. Annane et al. [2] reported that two $HBO_2$ sessions at 2.0 ATA were associated with worse outcomes than one $HBO_2$ session in comatose patients with acute CO poisoning, and that there was no evidence of the superiority of $HBO_2$ over $NBO_2$ in patients with transient loss of consciousness. Raphael et al. [3] reported that two of $HBO_2$ sessions at 2.0 ATA showed no beneficial effects versus one session in patients with CO poisoning who experienced sustained loss of consciousness. Further, one $HBO_2$ session was also ineffective versus $NBO_2$ therapy in patients who did not experience sustained loss of consciousness [3]. A recent meta-analysis of the therapeutic effects of different numbers of $HBO_2$ sessions found that $HBO_2$ therapy at a therapeutic pressure of 2.0 ATA was associated with a lower risk of memory impairment than $NBO_2$ therapy, but that two $HBO_2$ sessions was associated with a higher risk of memory impairment than one session [33]. However, as mentioned above, the therapeutic pressure of 2.0 ATA was considered to be

insufficient to produce its beneficial effects [19]. Therefore, multiple $HBO_2$ sessions with insufficient therapeutic pressure should be administered cautiously because of the possibility of worsening symptoms. However, the present data could not rule out the possibility that more severely affected patients had received more $HBO_2$ sessions because the $HBO_2$ therapy protocols were not consistent and depended on each institutions' policies [8].

In the present study, abnormal CT or MRI findings tended to be less frequent in the $NBO_2$ group than in the $HBO_2$ group, although the amount of data obtained was limited (Table 2). Previous studies have reported that imaging abnormalities are a risk factor for DNS [34, 35]. We could not include the abnormal CT or MRI findings as a variable in the logistic regression analyses to identify factors associated with the incidence of DNS and unimproved PCD because of the limited amount of data obtained in this study. However, the lower incidence of abnormal CT or MRI findings in the $NBO_2$ group than in the $HBO_2$ group may suggest that the patients in the $HBO_2$ group were more severely affected than those in the $NBO_2$ group. Myocardial injury is also associated with increased long-term mortality after CO poisoning [36]. In the present study, all the data related to myocardial injury, including the creatine kinase MB fraction, troponin T, and ECG abnormalities, tended to be worse in the $NBO_2$ group than in the $HBO_2$ group, although the amount of data was limited (Table 2). These results were inconsistent with the incidence of abnormal in CT or MRI findings. Data, including imaging findings and myocardial injury, were missing for some patients; therefore, it was unclear whether the severity differed between the two groups.

The number of patients treated with mechanical ventilation was significantly higher in the $NBO_2$ group than in the $HBO_2$ group (Table 1) and $PaO_2$ on arrival was significantly lower in the $NBO_2$ group than in the $HBO_2$ group (Table 3). This might have been related to the greater number of patients affected by fire in the $NBO_2$ group (Table 1). Patients affected by fires were more likely to suffer from smoke inhalation, and subsequently require intubation and ventilation because of their low $PaO_2$/fraction of inspiratory oxygen ratio. Intubated patients could not be treated with $HBO_2$ in a monoplace chamber, which may explain the large number of mechanically ventilated patients in the $NBO_2$ group. It was reported that fire causes cyanide poisoning concurrently with CO poisoning [37]. This might also be associated with the higher lactate levels in the $NBO_2$ group. More mechanically ventilated patients in the $NBO_2$ group also experienced longer ICU stays in the $NBO_2$ group. However, a sub-analysis after excluding mechanically ventilated patients yielded the same result, as $HBO_2$ therapy offered no advantage over $NBO_2$ therapy in the prevention of DNS, and multiple $HBO_2$ sessions on day 1 were still associated with a greater incidence of DNS.

Some retrospective studies have found that $HBO_2$ therapy has beneficial effects on the survival rate [38, 39] or activities of daily living (ADL) in patients with CO poisoning [40]. Rose et al. reported that $HBO_2$ therapy was associated with reduced in-hospital mortality and reduced 1-year mortality [38] and Huang et al. reported a lower 4-year mortality rate after treatment for CO poisoning [39]. In the present study, among 311 patients, there were three cases of CPAOA and three in-hospital deaths, but there were no deaths during the follow-up periods, although 41 patients were lost to follow-up. Regarding the effect of $HBO_2$ therapy on the survival rate after CO poisoning, our data did not reveal any evidence to support the previous reports [38, 39] because the follow-up period was only 2 months and 41 patients were lost to follow-up. Nakajima et al. reported that $HBO_2$ therapy was associated with a favorable consciousness level and ADL at discharge in patients with CO poisoning [40]. In the present study, cognitive dysfunction was only checked for 2 months after CO poisoning and there was no significant difference between the $NBO_2$ group and the $HBO_2$ group (Table 3). Therefore, further investigation is needed to explore the long-term beneficial effects of $HBO_2$ therapy.

There were several limitations to the present study. First, it was an observational study. Although there was no significant difference in the severity of poisoning among the subgroups

defined by the number of HBO$_2$ sessions received during first 24 h, the more severely affected patients, as assessed by the clinicians, may have received more HBO$_2$ sessions during the first 24 h. Second, the protocols for HBO$_2$ therapy, including the treatment pressure, number of sessions, their timing, and their duration, were not consistent. Third, as mentioned above, an equality of the groups was not maintained in some parts because this was an observational study. Fourth, there might be some selection bias because only 44% of patients in the NBO$_2$ group were transferred to hospitals where HBO$_2$ therapy was available. In those patients, mild cases might have received NBO$_2$ therapy, although the COHb levels in the NBO$_2$ group were not significantly different between patients transferred to institutions that only offered NBO$_2$ therapy (20.9 ± 10.0%) and patients transferred to institutions that also provided HBO$_2$ therapy (17.2 ± 10.1%, $P$ = 0.101). Furthermore, there may have been a selection bias on the part of the EMS, which may have sent less severely affected patients to institutions that only offered NBO$_2$ therapy. Finally, 40% of the patients without DNS were only diagnosed by telephone, so patients with mild symptoms might have been overlooked.

## Shortcomings

As mentioned above, there were some selection biases for non-randomized observational studies. In addition, the lack of a protocol for HBO$_2$ treatment made it difficult to interpret results such as dose-response between the number of HBO$_2$ treatments and the incidence of DNS. There were several issues with assessing DNS, including non-blinded evaluators, 13.8% of loss of follow-up, and the possibility of oversight of patients with mild symptoms.

## Conclusions

The practical clinical treatment for acute CO poisoning, including HBO$_2$ therapy, varied among the institutions participating in Japan. HBO$_2$ therapy with inconsistent protocols showed no advantages over NBO$_2$ therapy in the prevention of DNS or the improvement in PCD after CO poisoning. Furthermore, multiple HBO$_2$ sessions on the first day of hospitalization were associated with a greater incidence of DNS. Further research is required to clarify the efficacy of HBO$_2$ therapy in preventing DNS after CO poisoning.

## Supporting information

**S1 Data set.**
(CSV)

## Acknowledgments

**Members of the COP-J Study Investigators are given below:** Yamaguchi University Hospital, Ube (RT, MF, MT, KK), Kameda Medical Center, Kamogawa (SS), Tokai University School of Medicine, Isehara (SW), Hyogo Emergency Medical Center, Kobe (SK), Japanese Red Cross Maebashi Hospital, Maebashi (SS), Chiba University Graduate School of Medicine, Chiba (NH), Tokyo Medical and Dental University, Tokyo (KY), Japan Self Defense Forces Hospital Yokosuka, Yokosuka (KK), Hokkaido University, Sapporo (Tomonao Yoshida), Gunma University Hospital, Maebashi (Hiroaki Matsuoka), Kagawa University, Kagawa (Kenya Kawakita), Saiseikai Kumamoto Hospital, Kumamoto (Tadashi Kikuchi), Hiroshima University Hospital, Hiroshima (Satoshi Yamaga), St.Mary's Hospital, Kurume (Kazuhito Tamehiro), Tajima Emergency and Critical Care Medical Center, Toyooka Hospital, Toyooka (Osamu Fujisaki), Asahikawa Medical University Hospital, Asahikawa (Yuka Eto), Iwate Medical University, Morioka (Makoto Onodera), Ibaraki Prefectural Central Hospital, Kasama (Yoshimoto

Seki), Tokushima Red Cross Hospital, Komatsushima (Yasushi Fukuta), Kumamoto Red Cross Hospital, Kumamoto (Ken Kuwahara), Showa University, Tokyo (Kenichiro Fukuda), Nagano Red Cross Hospital, Nagano (Koji Yamakawa), Osaka University Graduate School of Medicine, Suita (Ryosuke Takegawa), Gifu University Graduate School of Medicine, Gifu (Tomoaki Doi), Yokohama City University Medical Center, Yokohama (Takuma Sakai), St. Luke's International Hospital, Tokyo (Shutaro Isokawa), Kanmon Medical Center, Shimonoseki (Shinichiro Tanaka), Tokuyama Central Hospital, Shunan (Susumu Yamashita), Kawasaki Medical School, Kurashiki (Yasukazu Shiino), Kumamoto University Hospital, Kumamoto (Tadashi Kaneko), Jichi Medical University, Tochigi (Chikara Yonekawa), National Hospital Organization Kumamoto Medical Center, Kumamoto (Masahiro Harada), Kindai University, Osaka (Takami Nakao), Tamaki Hospital, Hagi (Hideki Tamaki), Almeida Memorial Hospital, Oita (Nobuhiro Inagaki), Kanazawa University Hospital, Kanazawa (Masaki Okajima), Kagoshima University Graduate School of Medical and Dental Sciences, Kagoshima (Yasuyuki Kakihana), Aso Iizuka Hospital, Iizuka (Hiroshi Adachi), Nagasaki University Hospital, Nagasaki (Tomohito Hirao), Hiroshima Prefectural Hospital, Hiroshima (Masahiko Iseki), Saiseikai Matsuyama Hospital, Matsuyama (Katsusuke Kusunoki), Yamaguchi Prefectural Grand Medical Center, Hofu (Takeshi Inoue), Kurashiki Central Hospital, Kurashiki (Shinichiro Ienaga), Saiseikai Yamaguchi Hospital, Yamaguchi (Takashi Tamura), St Marianna University School of Medicine, Kawasaki (Nobuhiko Simozawa), Kochi Health Science Center, Kochi (Go Nojima), University Hospital of the Ryukyus, Okinawa (Kiyotaka Kohshi), Kohsei General Hospital, Mihara (Kenjiro Fujiwara), Kizawa Memorial Hospital, Minokamo (Mikito Yamada), Kagawa Rosai Hospital, Marugame (Kimihiro Yoshino), Osaka National Hospital, National Hospital Organization, Osaka (Daikai Sadamitsu), National Hospital Organization Medical Center, Tachikawa (Takashi Kanemura), Jichi Children's Medical Center Tochigi, Shimotsuke (Hidetaka Iwai), Ina Central Hospital, Ina (Aya Hori).

## Author Contributions

**Conceptualization:** Motoki Fujita, Masaki Todani, Kotaro Kaneda, Ryosuke Tsuruta.

**Data curation:** Motoki Fujita, Masaki Todani, Kotaro Kaneda, Shinya Suzuki, Shinjiro Wakai, Shota Kikuta, Satomi Sasaki, Noriyuki Hattori, Kazuyoshi Yagishita, Koji Kuwata, Ryosuke Tsuruta.

**Formal analysis:** Motoki Fujita.

**Investigation:** Motoki Fujita, Masaki Todani, Kotaro Kaneda, Shinya Suzuki, Shinjiro Wakai, Shota Kikuta, Satomi Sasaki, Noriyuki Hattori, Kazuyoshi Yagishita, Koji Kuwata, Ryosuke Tsuruta.

**Methodology:** Motoki Fujita, Masaki Todani, Kotaro Kaneda, Ryosuke Tsuruta.

**Project administration:** Motoki Fujita, Masaki Todani, Kotaro Kaneda, Ryosuke Tsuruta.

**Supervision:** Shinya Suzuki, Ryosuke Tsuruta.

**Validation:** Motoki Fujita, Masaki Todani, Kotaro Kaneda.

**Visualization:** Motoki Fujita.

**Writing – original draft:** Motoki Fujita.

**Writing – review & editing:** Masaki Todani, Kotaro Kaneda, Shinya Suzuki, Shinjiro Wakai, Shota Kikuta, Satomi Sasaki, Noriyuki Hattori, Kazuyoshi Yagishita, Koji Kuwata, Ryosuke Tsuruta.

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
