## [Decision Letter · Decision Letter 0]

6 Jan 2021

PONE-D-20-36680

Use of hyperbaric oxygen therapy for preventing delayed neurological sequelae in patients with carbon monoxide poisoning: a multicenter, prospective, observational study in Japan

PLOS ONE

Dear Dr. Fujita,

Thank you for submitting your manuscript to PLOS ONE. After careful consideration, we feel that it has merit but does not fully meet PLOS ONE’s publication criteria as it currently stands. Therefore, we invite you to submit a revised version of the manuscript that addresses the points raised during the review process.

We look forward to receiving your revised manuscript.

Kind regards,

Tai-Heng Chen, M.D.

Academic Editor

PLOS ONE

Reviewers' comments:

Reviewer's Responses to Questions

**Comments to the Author**

1. Is the manuscript technically sound, and do the data support the conclusions?

Reviewer #1: No

Reviewer #2: Yes

2. Has the statistical analysis been performed appropriately and rigorously? 

Reviewer #1: I Don't Know

Reviewer #2: Yes

3. Have the authors made all data underlying the findings in their manuscript fully available?

Reviewer #1: Yes

Reviewer #2: Yes

4. Is the manuscript presented in an intelligible fashion and written in standard English?

Reviewer #1: Yes

Reviewer #2: Yes

5. Review Comments to the Author

Reviewer #1: The study by Dr. Fujita and his team is a case series that included 255 patients with CO poisoning of a total 311 initially registered. In the group of 255, 171 received hyperbaric oxygen (HBO2) 1-3 times in the first 24 hours; the remaining patients (84) received normobaric oxygen (NBO2). Delayed neurological sequelae (DNS) occurred in 7.6% of the HBO2 group and 3.6% of the NBO2 group. The authors conclude that HBO2 therapy has no advantage over NBO2 therapy for the prevention of DNS or improving prolonged impaired consciousness. Interestingly, a factor associated with the incidence of DNS in this study is the number of HBO2 sessions in the first 24 h, suggesting that more HBO2 may worsen outcome. The finding of this retrospective analysis is surprising given the opposite conclusion of several randomized, blinded studies with HBO2 administered at what are believed to be therapeutic doses (2.4-2.8 ATA). If correct, the current paper could be an important finding that may affect current practice.

This is a moderately large case series, of the size needed to address this important question, and the paper is easy to read. There are a few minor points that need to be addressed. In addition, the data do not adequately support the authors’ conclusions.

The authors have made an effort to elaborate the patients’ initial conditions. However there are important differences in the two groups that may have affected outcomes. For example, there were differences in the percentages of patients who lost consciousness (42.3% vs. 48.0%, NBO2 vs. HBO2), SpCO at the scene (26.4% vs. 30.1%). These differences were not statistically significant but that does not exclude a possible effect on outcome. In this case the authors used multivariable logistic regression to sort that our, however they need to state how they chose the variables that were in their regression model. It would be helpful to have a table that includes both univariable and multivariable risk factors for outcome.

Another important factor that suggests selection bias is that only 44% of patients in the NBO2 group were transferred to hospitals where HBO2 was available. It is therefore plausible that initial evaluation at those hospitals triaged to NBO2 those patients who were a priori expected to do well without HBO2, thus enhancing the outcome of the NBO2 group. It is also conceivably there may have been a selection bias on the part of the EMS to send less severely affected patients to NBO2 hospitals. Please could the authors comment. Among those patients who were referred to a hospital with hyperbaric facilities, it would be useful for information on the indications for treatment in that subgroup.

The fact that SpCO wat the scene was 26.4% (NBO2) and 30.1% (HBO2), yet despite long times between exposure and hospital arrival (average of around 4 half times if the patients were breathing oxygen), COHb was decreased only to 19.3% and 18.7% in the two groups. Please could the authors comment.

In addition for the potential for selection bias, it is also possible that the relatively low numbers of adverse outcomes indicates that the study is underpowered to detect a difference. DNS was assessed either during a hospital visit or by phone. The relatively low rate of sequelae in this study compared with previous reports (e.g. Gorman et al, Anaesth Intensive Care. 1992;20:311-6, refs. 1 and 4) may have been because phone assessment is less sensitive than formal neuropsychological testing).

There were differences between groups regarding CT findings, which should be described, along with information on abnormal MRI findings. Previous studies have observed imaging abnormalities as risk factors for DNS (e.g. Jeon, et al. JAMA Neurol. 2018;75:436-443). It would be helpful to include an analysis of the subset of patients with CT or MRI abnormalities should be included.

In any retrospective analysis it is impossible to account for therapeutic decisions made based upon real-time assessment of patients, for example the observation that a greater number of HBO2 sessions in the first 24 hours was associated with a greater incidence of DNS. From this, the authors conclude: “Therefore, multiple rounds of HBO2 therapy should be administered with caution, because it is possible that symptoms will worsen.” Of the two factors here (number of treatments and clinical outcome), it is important to consider which one is cause and which one is effect. Contrary to the authors’ conclusion, it is more plausible that additional HBO2 was administered to patients who were not responding well.

The following is an overstatement: “Moreover, Scheinkestel et al. [6] reported harmful effects of HBO2 therapy in patients suffering acute CO poisoning.” It is also noteworthy that while this study reported fewer sequelae with normobaric vs. hyperbaric oxygen, if their protocol is to be the basis for treating CO poisoning, then 3-6 days of inpatient treatment are required.

The following statement, “Experimental data have shown that HBO2 induces oxidative stress in healthy rat brains, measured as the lipid peroxidation products in brain cortex homogenates [24–26]” should be amended to include the observation that in CO poisoning HBO2 actually reduces oxidative stress (Thom SR. Toxicol Appl Pharmacol 1993;123:248-56). It is also of note that other animal studies of CO poisoning have revealed other beneficial effects of HBO2, including inhibition of leukocyte beta-2 integrins (reference 28), reversal of CO-cytochrome c oxidase binding (Adv Exp Med Biol 1989;248:747-54) and recovery of energy metabolism (J Clin Invest 1992;89: 666-72).

The review of the literature in Conclusions suggests that all studies of HBO2 are equivalent, which they are not. Raphael’s and Annane’s studies were performed at 2 ATA, which as the authors point out is considered by many experts to be inadequate. In Raphael’s study the only comparison between NBO2 and HBO2 was in mild cases. Scheinkestel’s study utilized 3-6 days of therapy, which is usually considered impractical. On the other hand, studies at 2.5-2.8 ATA by Thom, Mathieu and Weaver have shown better outcome for HBO2. These details need to be included explicitly in the Discussion. The following statement pertains to a meta-analysis, “A recent meta-analysis of the therapeutic effects of different numbers of HBO2 sessions found that HBO2 therapy was associated with a lower risk of memory impairment than NBO2 therapy, but that two sessions of HBO2 therapy were associated with a higher risk of memory impairment than one session [30].” It is noteworthy that this meta-analysis also assumed equivalence of the various doses.

Reviewer #2: Thank you for allowing me to review this study by Motoki Fujita et al. I previously reviewed this manuscript for another journal. I feel like the manuscript is much improved from the prior version I reviewed.

The question of hyperbaric oxygen therapy efficacy to prevent delayed neurological sequelae in CO poisoning is very important. There has been much controversy over the topic. As the manuscript refers to, probably the best designed study - reported in Weaver et al NEJM 2002 - showed quite a positive benefit of using HBO2 in CO poisoning patients. When looking at the most up-to-date meta-analysis, however, combined, 6 studies show no benefit. The most recent one - performed in 2011 - actually showed some trend towards harm.

Regardless of the stance on HBO2 therapy effectiveness, even presuming it is effective, time delays to therapy (>5 hours in the Weaver study) could certainly limit the benefit of the therapy. Many centers do not offer emergent HBO2. Further studies in this situation are critical.

The authors present a prospective, multi-center clinical study on the use of HBO2 in acutely CO poisoned patients. This is valuable and a fairly large size.

There are some major limitations to the study in its design that could limit further applicability – this was not a randomized trial, there was not set HBO2 protocol (and many patients received non standard therapy previously tested it appears), and the definition of neurocognitive deficits was a bit murky. That said, this was an interesting study that is important to publish.

I am going to reveal my prior critiques and comment on responsiveness to those. I have some additional comments based on the new manuscript.

Prior review:

Major Issues:

1) How did providers determine who was to receive HBO2 therapy? This is not specified. I think if it was up to the provider, more analysis of the decisions/trends of the patients need to be identified. Did they use a general criteria for the trial? Did they go with a professional organization recommendation? The American College of Emergency Physicians does not provide specific recommendations, of note.

-> There was quite a bit of discussion around how many patients received what therapy. It was quite non standard but it was sufficiently reported here. Some were very strange (15 HBO2 sessions over a week), so not sure how to interpret these.

2) Mechanically ventilated patients were not included in the analysis. This is a major limitation. By my calculation, only 4.6% of the HBO2 patients required mechanical ventilation, whereas, 19.5% of the NBO2 group required mechanical ventilation. These patients were then not analyzed. This is a major limitation as these would be the most severely ill patients. Ideally, there is some sensitivity analysis that would include these patients. This may be why the delayed neurological sequelae are so low in the NBO2 group - the sickest were eliminated (19.5%). I do agree to not include patients that had an arrest upon arrival for this study.

-> The authors included mechanical ventilation patients in the analysis. This was a great improvement in the study and makes it much more applicable.

3) The definition of DNS is very vague. Page 7 refers to patients were "checked by a physician if any doubt". What does this mean? This needs to be listed very specifically how DNS was determined as it is a primary outcome of the study. If some of the list or discussion of the DNS determination algorithm needs to be in supplemental materials, that is fine. The exact method needs to be reported.

-> Clearer algorithm certainly. Still some limits. Is the screening questions asked ever been standardized or validated previously? With the 4x mental status exams, did the patient need to have an abnormality on any one of these or multiple tests? If it is only one deficit on a single test this could be a biased definition. The opposite is true as well – if very strict on defining (for instance, patients had to have all 4 with a deficit, then could be biased the other way (not reporting the DNS enough). I would clarify. This is much improved already though.

4) No reports on mortality in this group. I think it would be helpful. The size of the study and exclusion of cardiac arrest upon arrival patients does not warrant this being an outcome measure; however, the numbers should be reported.

-> better commentary on mortality overall

5) Exposure distribution was unique - the most common causes of CO poisoning are generally motor vehicle exhaust and fires. Here, >50% of patients were exposed to burning charcoal. Is this the most common form of poisoning in Japan? This should be discussed more. Related, a lot more of the NBO2 group was exposed to fire, generally felt as the most severe form of exposure. This would provide that the effect seen is even more surprising.

-> much clearer. I would use term “intentional” not “suicide”. For the statistical comparison, I would do statistics for each exposure type (fire, exhaust, coal) instead of exposure in general.

6) Why were the DNS events so low. Most reported prevalence is between 20-40% long term. Why was there only one DNS event in the NBO2 group? Is this because mechanical ventilated patients are excluded from the analysis? Again, I think this data needs to be reported at least in a sensitivity analysis if available. Else, should be listed as a major limitation.

-> Now addressed mechanical ventilation issue. Still quite low though. Any reason why you suspect? Perhaps points above about definition of DNS would be helpful to learn.

7) Why was the PaO2 higher in the HBO2 group? When was this lab taken? Was it prior to HBO2 initiation?

-> This was discussed a bit. Not sure why it is higher but not sure clinical meaningfulness.

8) The issue of heterogeneity of therapy needs to be addressed as a major limitation - no standard intervention, multiple different pressures/number of hyperbaric treatments for the patients.

-> this was well discussed.

Applicable Minor issues:

4) 35% of facilities did not have access to HBO2 - this is actually a high percentage of HBO2 capable facilities. The US only has 250-350 nationwide. Is there a lot more hyperbaric centers in Japan or is it more that centers with HBO2 were more likely to enroll in this study? Is there data for how many centers in Japan?

-> related, I would include how many patients required intrafacility transfer for HBO2.

6) Page 10, what does "period of oxygen administration" mean? I am confused by this term. Does that mean time before presentation to hospital on oxygen?

-> I am still not sure about this term. It is clearer how it is used now, but I think general length of stay in hospital is more meaningful. Could probably drop it. We can presume most of the patients received oxygen I think.

7) When discussing controversy around HBO2 being effective or not, I would include the discussion of some of the large retrospective studies that suggest mortality or ADL benefit to HBO2:

a. Rose JJ, Nouraie M, Gauthier MC, et al. Clinical Outcomes and Mortality Impact of Hyperbaric Oxygen Therapy in Patients With Carbon Monoxide Poisoning. Crit Care Med. 2018;46(7):e649-e655. doi:10.1097/CCM.0000000000003135

b. Nakajima M, Aso S, Matsui H, Fushimi K, Yasunaga H. Hyperbaric oxygen therapy and mortality from carbon monoxide poisoning: A nationwide observational study. Am J Emerg Med. 2020;38(2):225-230. doi:10.1016/j.ajem.2019.02.009

-> I would still include this

Additional comments:

1) CK-MB is used for cardiac involvement. What was your cut off? How many had “High” values? Do you have troponin data available?

2) What were the ECG abnormalities seen? 20-22% incidence is quite high.

3) What were the CT abnormalities seen? Head CT or Chest CT?

4) What were MRI abnormalities seen, quite a few in the HBO2 group.

5) Not sure you need a paragraph describing vital signs, we can read table

6) Why was there bed rest prescribed and what is the meaning of it? I did not think we prescribe this in clinical care anymore very much.

7) Give % mortality of overall cohort (included and excluded) when you report the number I think.

6. PLOS authors have the option to publish the peer review history of their article (what does this mean?). If published, this will include your full peer review and any attached files.

Reviewer #1: **Yes: **Richard Moon

Reviewer #2: No

---

## [Author Response · Author response to Decision Letter 0]

21 Apr 2021

We are grateful to the reviewers for their critical comments and useful suggestions, which have helped us to improve our paper considerably. As indicated in the responses that follow, we have taken all these comments and suggestions into account in the revised version of our paper. In the manuscript, the revised text is shown in red font. We have also provided the page and line numbers corresponding to the revisions in the text.

Reviewer 1

1．The authors have made an effort to elaborate the patients’ initial conditions. However, there are important differences in the two groups that may have affected outcomes. For example, there were differences in the percentages of patients who lost consciousness (42.3% vs. 48.0%, NBO2 vs. HBO2), SpCO at the scene (26.4% vs. 30.1%). These differences were not statistically significant but that does not exclude a possible effect on outcome. In this case the authors used multivariable logistic regression to sort that our, however they need to state how they chose the variables that were in their regression model. It would be helpful to have a table that includes both univariable and multivariable risk factors for outcome.

Response: 

We have added the results of the univariate regression analyses of all the factors to Tables 5 and 6. The factors associated with DNS and unimproved PCD in previous reports [12–17] were included as variables in the multivariable logistic regression models. Because of the small numbers of patients whose SpCO at the scene (n = 8 in the NBO2 group and n = 40 in the HBO2 group; Table 1), SpCO at the scene was not included in the univariate or multivariable logistic regression analyses.

2. Another important factor that suggests selection bias is that only 44% of patients in the NBO2 group were transferred to hospitals where HBO2 was available. It is therefore plausible that initial evaluation at those hospitals triaged to NBO2 those patients who were a priori expected to do well without HBO2, thus enhancing the outcome of the NBO2 group. It is also conceivably there may have been a selection bias on the part of the EMS to send less severely affected patients to NBO2 hospitals. Please could the authors comment. Among those patients who were referred to a hospital with hyperbaric facilities, it would be useful for information on the indications for treatment in that subgroup.

Response: 

There may be a selection bias because only 44% of patients in the NBO2 group were transferred to hospitals where HBO2 therapy was available. Among these, mildly affected patients may have received NBO2 therapy, although there was no significant difference in the COHb levels of the NBO2 group between patents transferred to institutions that offered only NBO2 therapy and patients transferred to institutions that also provided HBO2 therapy (20.9 ± 10.0% and 17.2 ± 10.1%, respectively, P = 0.101). There may be a selection bias arising from the EMS, as you mentioned, although some emergency centers did not have HBO2 chambers. We have added an explanation of this to the limitations (page 21, line 414) and the COHb levels to the results (page 9, line 187).

3. The fact that SpCO wat the scene was 26.4% (NBO2) and 30.1% (HBO2), yet despite long times between exposure and hospital arrival (average of around 4 half times if the patients were breathing oxygen), COHb was decreased only to 19.3% and 18.7% in the two groups. Please could the authors comment.

Response: 

In our study, the numbers of patients with SpCO data were low (n = 8 in the NBO2 group and n = 40 in the HBO2 group; Table 1). This may help explain why COHb did not decrease enough on admission.

4. In addition for the potential for selection bias, it is also possible that the relatively low numbers of adverse outcomes indicates that the study is underpowered to detect a difference. DNS was assessed either during a hospital visit or by phone. The relatively low rate of sequelae in this study compared with previous reports (e.g. Gorman et al, Anaesth Intensive Care. 1992;20:311-6, refs. 1 and 4) may have been because phone assessment is less sensitive than formal neuropsychological testing.

Response: 

We had previously described this limitation in the discussion section. We have added more text regarding this point in the discussion (page 16, line 300). 

5. There were differences between groups regarding CT findings, which should be described, along with information on abnormal MRI findings. Previous studies have observed imaging abnormalities as risk factors for DNS (e.g. Jeon, et al. JAMA Neurol. 2018;75:436-443). It would be helpful to include an analysis of the subset of patients with CT or MRI abnormalities should be included.

Response: 

We have added more text to the discussion describing the abnormal CT or MRI findings (page 18, line 360). However, because the amount of data was limited, we could not include the incidence of abnormal CT or MRI findings as a variable in the multivariable logistic regression models to identify factors associated with the incidence of DNS and unimproved PCD.

6. In any retrospective analysis it is impossible to account for therapeutic decisions made based upon real-time assessment of patients, for example the observation that a greater number of HBO2 sessions in the first 24 hours was associated with a greater incidence of DNS. From this, the authors conclude: “Therefore, multiple rounds of HBO2 therapy should be administered with caution, because it is possible that symptoms will worsen.” Of the two factors here (number of treatments and clinical outcome), it is important to consider which one is cause and which one is effect. Contrary to the authors’ conclusion, it is more plausible that additional HBO2 was administered to patients who were not responding well.

Response: 

We have added text regarding this point to the discussion (page 18, line 355).

7. The following is an overstatement: “Moreover, Scheinkestel et al. [6] reported harmful effects of HBO2 therapy in patients suffering acute CO poisoning.” It is also noteworthy that while this study reported fewer sequelae with normobaric vs. hyperbaric oxygen, if their protocol is to be the basis for treating CO poisoning, then 3-6 days of inpatient treatment are required.

Response: 

We have deleted this sentence.

8. The following statement, “Experimental data have shown that HBO2 induces oxidative stress in healthy rat brains, measured as the lipid peroxidation products in brain cortex homogenates [24–26]” should be amended to include the observation that in CO poisoning HBO2 actually reduces oxidative stress (Thom SR. Toxicol Appl Pharmacol 1993;123:248-56). It is also of note that other animal studies of CO poisoning have revealed other beneficial effects of HBO2, including inhibition of leukocyte beta-2 integrins (reference 28), reversal of CO-cytochrome c oxidase binding (Adv Exp Med Biol 1989;248:747-54) and recovery of energy metabolism (J Clin Invest 1992;89: 666-72).

Response: 

We have added an explanation to the discussion and have reordered the discussion (page 17, line 225).

9. The review of the literature in Conclusions suggests that all studies of HBO2 are equivalent, which they are not. Raphael’s and Annane’s studies were performed at 2 ATA, which as the authors point out is considered by many experts to be inadequate. In Raphael’s study the only comparison between NBO2 and HBO2 was in mild cases. Scheinkestel’s study utilized 3-6 days of therapy, which is usually considered impractical. On the other hand, studies at 2.5-2.8 ATA by Thom, Mathieu and Weaver have shown better outcome for HBO2. These details need to be included explicitly in the Discussion. The following statement pertains to a meta-analysis, “A recent meta-analysis of the therapeutic effects of different numbers of HBO2 sessions found that HBO2 therapy was associated with a lower risk of memory impairment than NBO2 therapy, but that two sessions of HBO2 therapy were associated with a higher risk of memory impairment than one session [30].” It is noteworthy that this meta-analysis also assumed equivalence of the various doses.

Response:

The difference in the therapeutic pressure of the HBO2 therapy has been added to the discussion (page 18, lines 341–343, 347, and 351).

 

We are grateful to the reviewers for their critical comments and useful suggestions, which have helped us to improve our paper considerably. As indicated in the responses that follow, we have taken all these comments and suggestions into account in the revised version of our paper. In the manuscript, the revised text is shown in red. We have also provided the page and line numbers corresponding to the revisions in the text.

Reviewer 2

Prior review: Major Issues:

1. How did providers determine who was to receive HBO2 therapy? This is not specified. I think if it was up to the provider, more analysis of the decisions/trends of the patients need to be identified. Did they use a general criteria for the trial? Did they go with a professional organization recommendation? The American College of Emergency Physicians does not provide specific recommendations, of note.

-> There was quite a bit of discussion around how many patients received what therapy. It was quite non standard but it was sufficiently reported here. Some were very strange (15 HBO2 sessions over a week), so not sure how to interpret these.

Response: 

The HBO2 therapy protocols were not consistent and depended on the practices at each participating institution, as we reported in reference 8.

2. Mechanically ventilated patients were not included in the analysis. This is a major limitation. By my calculation, only 4.6% of the HBO2 patients required mechanical ventilation, whereas, 19.5% of the NBO2 group required mechanical ventilation. These patients were then not analyzed. This is a major limitation as these would be the most severely ill patients. Ideally, there is some sensitivity analysis that would include these patients. This may be why the delayed neurological sequelae are so low in the NBO2 group - the sickest were eliminated (19.5%). I do agree to not include patients that had an arrest upon arrival for this study.

-> The authors included mechanical ventilation patients in the analysis. This was a great improvement in the study and makes it much more applicable.

Response: 

Data about mechanically ventilated patients were added to manuscript, as you suggested.

3. The definition of DNS is very vague. Page 7 refers to patients were "checked by a physician if any doubt". What does this mean? This needs to be listed very specifically how DNS was determined as it is a primary outcome of the study. If some of the list or discussion of the DNS determination algorithm needs to be in supplemental materials, that is fine. The exact method needs to be reported.

-> Clearer algorithm certainly. Still some limits. Is the screening questions asked ever been standardized or validated previously? With the 4x mental status exams, did the patient need to have an abnormality on any one of these or multiple tests? If it is only one deficit on a single test this could be a biased definition. The opposite is true as well – if very strict on defining (for instance, patients had to have all 4 with a deficit, then could be biased the other way (not reporting the DNS enough). I would clarify. This is much improved already though.

Response: 

DNS was diagnosed by a physician based on all the findings at the time of diagnosis, including a cognitive function test, such as the mini-mental state examination, the Wechsler adult intelligence scale, Hasegawa’s dementia scale-revised, the trail-making test, or the story recall test. We have revised the text in the data collection and analysis section (page 5, line 119).

4. No reports on mortality in this group. I think it would be helpful. The size of the study and exclusion of cardiac arrest upon arrival patients does not warrant this being an outcome measure; however, the numbers should be reported.

-> better commentary on mortality overall

Response: 

In the present study, among 311 patients, there were three cases of CPAOA and three in-hospital deaths, but there were no patient deaths recorded during the follow-up period. However, 41 patients were lost to follow-up. We have added this to the discussion (page 20, line 391).

5. Exposure distribution was unique - the most common causes of CO poisoning are generally motor vehicle exhaust and fires. Here, >50% of patients were exposed to burning charcoal. Is this the most common form of poisoning in Japan? This should be discussed more. Related, a lot more of the NBO2 group was exposed to fire, generally felt as the most severe form of exposure. This would provide that the effect seen is even more surprising.

-> much clearer. I would use term “intentional” not “suicide”. For the statistical comparison, I would do statistics for each exposure type (fire, exhaust, coal) instead of exposure in general.

Response: 

In previous reports, the most common causes of CO poisoning were internal combustion engines (Weaver et al., NEJM 2002;347: 1057-1067.) or gas water heaters (Annane et al., Intensive Care Med. 2011;3: 486-492). However, it was also reported that charcoal is a major cause of CO poisoning in Asia (Lin et al. J Chin Med Assoc. 2018;81: 682-690; Ku et al. Gen Hosp Psychiatry. 2010;32: 310-4.). Charcoal is still used for heating in northern Japan. We have also replaced the term “suicide” to “intentional” (page 8, Table 1) and have added data for each exposure type (fire, exhaust, coal) to Table 1.

6. Why were the DNS events so low. Most reported prevalence is between 20-40% long term. Why was there only one DNS event in the NBO2 group? Is this because mechanical ventilated patients are excluded from the analysis? Again, I think this data needs to be reported at least in a sensitivity analysis if available. Else, should be listed as a major limitation.

-> Now addressed mechanical ventilation issue. Still quite low though. Any reason why you suspect? Perhaps points above about definition of DNS would be helpful to learn.

Response: 

The definition of DNS may partly explain why its incidence was low in this study. We have added a sentence to the discussion to describe this point (page 16, line 300).

7. Why was the PaO2 higher in the HBO2 group? When was this lab taken? Was it prior to HBO2 initiation?

-> This was discussed a bit. Not sure why it is higher but not sure clinical meaningfulness.

Response: 

This might have been related to the greater number of patients affected by fires in the NBO2 group (Table 1). Patients affected by fires were more likely to suffer from smoke inhalation, and subsequently require intubation and ventilation (Table 3) because of a low P/F ratio. We have added an explanation of this to the discussion (page 119, lines 377 and 381). 

8. The issue of heterogeneity of therapy needs to be addressed as a major limitation - no standard intervention, multiple different pressures/number of hyperbaric treatments for the patients.

-> this was well discussed.

Response: 

We have added these points to the limitations (page 21, line 414).

Applicable Minor issues:

1. 35% of facilities did not have access to HBO2 - this is actually a high percentage of HBO2 capable facilities. The US only has 250-350 nationwide. Is there a lot more hyperbaric centers in Japan or is it more that centers with HBO2 were more likely to enroll in this study? Is there data for how many centers in Japan?

-> related, I would include how many patients required intrafacility transfer for HBO2.

Response: 

At the start of the study, there were 568 institutions in Japan that had an HBO2 chamber, of which 115 institutions had a board-certified fellow of the Japanese Society of Hyperbaric and Undersea Medicine. We have described this in the methods section (page 4, line 98).

Unfortunately, we have no data about interfacility transfer for HBO2 therapy.

2. Page 10, what does "period of oxygen administration" mean? I am confused by this term. Does that mean time before presentation to hospital on oxygen?

-> I am still not sure about this term. It is clearer how it is used now, but I think general length of stay in hospital is more meaningful. Could probably drop it. We can presume most of the patients received oxygen I think.

Response: 

This term refers to the period of oxygen administration during hospital stay; oxygen is not always administered for the entire hospital stay.

3. When discussing controversy around HBO2 being effective or not, I would include the discussion of some of the large retrospective studies that suggest mortality or ADL benefit to HBO2:

a. Rose JJ, Nouraie M, Gauthier MC, et al. Clinical Outcomes and Mortality Impact of Hyperbaric Oxygen Therapy in Patients With Carbon Monoxide Poisoning. Crit Care Med. 2018;46(7):e649-e655. doi:10.1097/CCM.0000000000003135

b. Nakajima M, Aso S, Matsui H, Fushimi K, Yasunaga H. Hyperbaric oxygen therapy and mortality from carbon monoxide poisoning: A nationwide observational study. Am J Emerg Med. 2020;38(2):225-230. doi:10.1016/j.ajem.2019.02.009

-> I would still include this.

Response: 

We have escribed the survival rate and ADL of patients with CO poisoning in the discussion (page 20, line 391).

Additional comments:

1. CK-MB is used for cardiac involvement. What was your cut off? How many had “High” values? Do you have troponin data available?

Response: 

We have added the number of the patients whose CK-MB was above the normal range to Table 1.

2. What were the ECG abnormalities seen? 20-22% incidence is quite high.

Response: 

We have added the details of ECG abnormalities to Table 1.

3. What were the CT abnormalities seen? Head CT or Chest CT?

Response: 

We have added the details of CT abnormalities to Table 1.

4. What were MRI abnormalities seen, quite a few in the HBO2 group.

Response: 

We have added the details of MRI abnormalities to Table 1.

5. Not sure you need a paragraph describing vital signs, we can read table

Response: 

We have deleted the text about vital signs.

6. Why was there bed rest prescribed and what is the meaning of it? I did not think we prescribe this in clinical care anymore very much.

Response: 

We thought that bed rest might be related to the prevention of DNS. However, the length of bed rest seems to be affected by the period of mechanical ventilation in the present study. Therefore, we have deleted these data from Table 3.

7. Give % mortality of overall cohort (included and excluded) when you report the number I think.

Response:

We have included the number of deceased patients in the discussion, as you suggested (page 20, line 391).

---

## [Decision Letter · Decision Letter 1]

9 May 2021

PONE-D-20-36680R1

Use of hyperbaric oxygen therapy for preventing delayed neurological sequelae in patients with carbon monoxide poisoning: a multicenter, prospective, observational study in Japan

PLOS ONE

Dear Dr. Fujita,

Thank you for submitting your manuscript to PLOS ONE. After careful consideration, we feel that it has merit but does not fully meet PLOS ONE’s publication criteria as it currently stands. Therefore, we invite you to submit a revised version of the manuscript that addresses the points raised during the review process.

We look forward to receiving your revised manuscript.

Kind regards,

Tai-Heng Chen, M.D.

Academic Editor

PLOS ONE

Reviewers' comments:

Reviewer's Responses to Questions

**Comments to the Author**

1. If the authors have adequately addressed your comments raised in a previous round of review and you feel that this manuscript is now acceptable for publication, you may indicate that here to bypass the “Comments to the Author” section, enter your conflict of interest statement in the “Confidential to Editor” section, and submit your "Accept" recommendation.

Reviewer #1: (No Response)

2. Is the manuscript technically sound, and do the data support the conclusions?

Reviewer #1: No

3. Has the statistical analysis been performed appropriately and rigorously? 

Reviewer #1: I Don't Know

4. Have the authors made all data underlying the findings in their manuscript fully available?

Reviewer #1: Yes

5. Is the manuscript presented in an intelligible fashion and written in standard English?

Reviewer #1: Yes

6. Review Comments to the Author

Reviewer #1: Dr. Fujita and his team have responded appropriately to many of the reviewers’ comments, but several concerns remain.

Lines 325-338: the authors are attempting to make the point that HBO2-induced oxidative stress could the cause of worse outcome in the HBO2 group. In favor of this hypothesis they cite evidence that HBO2 induces oxidative stress. Of course it does. However, so does carbon monoxide exposure, and published evidence indicates that HBO2 reduces it (reference 26). The conclusion in lines 335-338 therefore needs to be amended.

Lines 350-354: “A recent meta-analysis of the therapeutic effects of different numbers of HBO2 sessions, including a different range of therapeutic pressures from 2.0 to 2.8 ATA, found that HBO2 therapy was associated with a lower risk of memory impairment than NBO2 therapy, but that two HBO2 sessions was associated with a higher risk of memory impairment than one session [33].” This particular conclusion of Wang’s meta-analysis is based upon two studies (Raphael and Annane) in which sub-therapeutic HBO2 doses were used, and both mild and moderate sequelae (including memory impairment) were defined only by a patient questionnaire (no formal testing of memory); only 4 patients out of 170 had severe sequelae defined as objective abnormalities. All studies using HBO2 at 2.0 ATA have shown no benefit, while all studies using HBO2 at 2.8-3.0 ATA have shown benefit. Please add a caveat to this discussion.

There were 171 patients in the HBO2 group, however the number who received a first HBO2 treatment (Table 2) totaled only 165. Please explain.

From the previous review: It would be helpful to include an analysis of the subset of patients with CT or MRI abnormalities. What were the outcomes for those individuals with brain abnormalities on either CT or MRI?

The following points need to be addressed in a ‘Shortcomings’ section after the Discussion.

(1). This is not a randomized study, therefore, as pointed out in the previous review, the possibility of selection bias is large, both at the EMS level and within hospitals where HBO2 was available. Indeed, there is evidence that those patients who were treated with hyperbaric oxygen (HBO2) were more severely affected than those who were not. In addition to differences pointed out in the previous review (percentages of patients who lost consciousness, SpCO at the scene), the percentage of patients with abnormal CT, basal ganglia lesions) and MRI (basal ganglia lesions) was higher in the HBO2 group. The lack of statistical significance does not exclude a clinical relationship, particularly since fewer than 60% of patients who received HBO2 in this series were treated at a pressure considered to be therapeutic (≥2.4 ATA).

(2). There is no evidence in the manuscript that the assessors were blinded as to the treatment group (NBO2 vs. HBO2).

(3). An apparent dose-response between number of HBO2 treatments and incidence of delayed neurological sequelae (DNS) is incorrectly interpreted as evidence of possible harm due to HBO2. There is no statistical way to control retrospectively for a clinical observation related to improvement/lack of improvement during therapy. For example, a study of seriously ill infected patients might reveal a relationship between duration of antibiotic therapy and mortality. However on this basis it would be incorrect to conclude that antibiotic therapy is harmful; rather, longer antibiotic therapy was most likely prescribed because of lower clinical response to therapy in more severe infections.

(4). The relatively low statistical power to exclude an effect given the low rate of DNS and the fact that that many patients were not formally tested (the numbers are not stated) and 16% were lost to follow-up.

7. PLOS authors have the option to publish the peer review history of their article (what does this mean?). If published, this will include your full peer review and any attached files.

Reviewer #1: **Yes: **Richard Moon

---

## [Author Response · Author response to Decision Letter 1]

19 May 2021

We are grateful to the reviewer 1 for his critical comments and useful suggestions, which have helped us to improve our paper considerably. As indicated in the responses that follow, we have taken all these comments and suggestions into account in the revised version of our paper. In the manuscript, the revised text is shown in red font. We have also provided the page and line numbers corresponding to the revisions in the text.

Reviewer 1

1. Lines 325-338: the authors are attempting to make the point that HBO2-induced oxidative stress could the cause of worse outcome in the HBO2 group. In favor of this hypothesis they cite evidence that HBO2 induces oxidative stress. Of course it does. However, so does carbon monoxide exposure, and published evidence indicates that HBO2 reduces it (reference 26). The conclusion in lines 335-338 therefore needs to be amended.

Response: 

We have changed the sentence as you mentioned (page 18, line344-346). 

2. Lines 350-354: “A recent meta-analysis of the therapeutic effects of different numbers of HBO2 sessions, including a different range of therapeutic pressures from 2.0 to 2.8 ATA, found that HBO2 therapy was associated with a lower risk of memory impairment than NBO2 therapy, but that two HBO2 sessions was associated with a higher risk of memory impairment than one session [33].” This particular conclusion of Wang’s meta-analysis is based upon two studies (Raphael and Annane) in which sub-therapeutic HBO2 doses were used, and both mild and moderate sequelae (including memory impairment) were defined only by a patient questionnaire (no formal testing of memory); only 4 patients out of 170 had severe sequelae defined as objective abnormalities. All studies using HBO2 at 2.0 ATA have shown no benefit, while all studies using HBO2 at 2.8-3.0 ATA have shown benefit. Please add a caveat to this discussion.

Response: 

We had already discussed the difference of therapeutic effects between 2.0 ATA and 2.5 to 3.0 ATA in the 3rd paragraph of the discussion (page16-17, line 315-333). We have changed the sentence of the discussion because our interpretation of the recent meta-analysis is incorrect as you mentioned (page 18, line 359-365). 

3. There were 171 patients in the HBO2 group, however the number who received a first HBO2 treatment (Table 2) totaled only 165. Please explain.

Response: 

In the six patients, HBO2 therapy were not administered during the first 24 h in 2 patients of the HBO2 group and the details of HBO2 therapy were unknown in 4 patents. This was the reason why the number of the patients received a first HBO2 treatment was 165. 

We have added the contents to the results sessions of the revised manuscript (page 10, line 211-213).

4. From the previous review: It would be helpful to include an analysis of the subset of patients with CT or MRI abnormalities. What were the outcomes for those individuals with brain abnormalities on either CT or MRI?

Response: 

Among 35 patients with abnormal findings in CT or MRI, DNS was observed in 2 (22.2%) and 8 (30.8%) patients in the NBO2 group (n = 9) and the HBO2 group (n=26), respectively. There was no significant difference in the incidence of DNS between the groups (P = 0.625). Unimproved PCD was observed in 2 (22.2%) and 6 (23.1%) patients in the NBO2 group and the HBO2 group, respectively. There was no significant difference between the groups (P = 0.958).

We have added the contents to the results sessions of the revised manuscript (page 13, line 252-257).

5. The following points need to be addressed in a ‘Shortcomings’ section after the Discussion.

(1). This is not a randomized study, therefore, as pointed out in the previous review, the possibility of selection bias is large, both at the EMS level and within hospitals where HBO2 was available. Indeed, there is evidence that those patients who were treated with hyperbaric oxygen (HBO2) were more severely affected than those who were not. In addition to differences pointed out in the previous review (percentages of patients who lost consciousness, SpCO at the scene), the percentage of patients with abnormal CT, basal ganglia lesions) and MRI (basal ganglia lesions) was higher in the HBO2 group. The lack of statistical significance does not exclude a clinical relationship, particularly since fewer than 60% of patients who received HBO2 in this series were treated at a pressure considered to be therapeutic (≥2.4 ATA).

(2). There is no evidence in the manuscript that the assessors were blinded as to the treatment group (NBO2 vs. HBO2).

(3). An apparent dose-response between number of HBO2 treatments and incidence of delayed neurological sequelae (DNS) is incorrectly interpreted as evidence of possible harm due to HBO2. There is no statistical way to control retrospectively for a clinical observation related to improvement/lack of improvement during therapy. For example, a study of seriously ill infected patients might reveal a relationship between duration of antibiotic therapy and mortality. However on this basis it would be incorrect to conclude that antibiotic therapy is harmful; rather, longer antibiotic therapy was most likely prescribed because of lower clinical response to therapy in more severe infections.

(4). The relatively low statistical power to exclude an effect given the low rate of DNS and the fact that that many patients were not formally tested (the numbers are not stated) and 16% were lost to follow-up.

Response: 

We have briefly added the ‘Shortcomings’ section after the Discussion because majority of the contents have already addressed as limitations (page 21-22, line 435-441). 

Further, we have added the sentence about non-blinded evaluator for DNS in the method session (page 5, line12-124).

---

## [Decision Letter · Decision Letter 2]

9 Jun 2021

Use of hyperbaric oxygen therapy for preventing delayed neurological sequelae in patients with carbon monoxide poisoning: a multicenter, prospective, observational study in Japan

PONE-D-20-36680R2

Dear Dr. Fujita,

We’re pleased to inform you that your manuscript has been judged scientifically suitable for publication and will be formally accepted for publication once it meets all outstanding technical requirements.

Kind regards,

Tai-Heng Chen, M.D.

Academic Editor

PLOS ONE

Reviewers' comments:

Reviewer's Responses to Questions

**Comments to the Author**

1. If the authors have adequately addressed your comments raised in a previous round of review and you feel that this manuscript is now acceptable for publication, you may indicate that here to bypass the “Comments to the Author” section, enter your conflict of interest statement in the “Confidential to Editor” section, and submit your "Accept" recommendation.

Reviewer #1: (No Response)

2. Is the manuscript technically sound, and do the data support the conclusions?

Reviewer #1: No

3. Has the statistical analysis been performed appropriately and rigorously? 

Reviewer #1: I Don't Know

4. Have the authors made all data underlying the findings in their manuscript fully available?

Reviewer #1: Yes

5. Is the manuscript presented in an intelligible fashion and written in standard English?

Reviewer #1: No

6. Review Comments to the Author

Reviewer #1: Dr. Fujita and colleagues have done a tremendous amount of work in this study, however modifications are still needed to communicate the differences between this retrospective study and a randomized, blinded prospective study.

The following paragraph has been edited by the authors, but in a way to focus on possible adverse effects of oxidative stress due to administration of hyperbaric oxygen for CO poisoning (lines 334-346):

“Oxidative stress is a key mechanism in DNS [20–25]…Although HBO2 therapy has beneficial effects, it should be considered that there are concerns about adverse effects of HBO2 therapy such as HBO2-induced oxidative stress.”

However, the biochemical data from animals and humans overwhelmingly point to a reduction in oxidative stress in CO poisoning after hyperbaric oxygen. A more appropriate wording of that paragraph would be:

“Oxidative stress is a key mechanism in DNS [20–25]. However, there have been reports that HBO2 therapy itself induces oxidative stress [29–32]. Experimental data have shown that HBO2 induces oxidative stress in healthy rat brains, measured as the lipid peroxidation products in brain cortex homogenates [29–31]. This HBO2-induced oxidative stress is related to the HBO2 pressure [29] or the exposure time [30]. It has also been reported that a 342 single session of HBO2 (2.4 kPa, 131 min) reduced plasma vitamin C and increased plasma lipid peroxides and urinary 8-oxo-deoxyguanosine excretion in healthy volunteers [32]. Although there are concerns about adverse effects of HBO2 therapy such as HBO2-induced oxidative stress, measurement of oxidative stress in animal models of CO poisoning has demonstrated reduced lipid peroxidation [26], inhibition of leukocyte beta-2 integrins [18], reversal of CO-cytochrome c oxidase binding [27] and recovery of energy metabolism [28]. It should also be noted that markers of oxidative stress in humans poisoned by carbon monoxide are reduced after treatment (https://pubmed.ncbi.nlm.nih.gov/21424975/).”

Lines 364-368 have been edited to address the issue of selection bias: “Therefore, multiple HBO2 sessions with insufficient therapeutic pressure should be administered cautiously because of the possibility of worsening symptoms. However, the present data could not rule out the possibility that more severely affected patients had received more HBO2 sessions because the HBO2 therapy protocols were not consistent and depended on each institutions’ policies [8].” The manuscript should also note that a randomized blinded study of three hyperbaric oxygen treatments vs. one revealed no evidence of increased adverse outcomes (page 155 of Weaver LK. Carbon monoxide poisoning. Undersea Hyperb Med 2020;47(1):151-69).

Lines 439-441: “There were several issues with assessing DNS, including non-blinded evaluators, 13.8% of loss of follow-up, and the possibility of oversight of patients with mild symptoms.” In the manuscript it is stated that 41 patients were lost to follow-up (lines 406, 409). The total number of patients is 255 (Table 1), thus percentage lost is not 13.8% but 16.1%.

The shortcomings section should be formatted per the prior review. Although some of the shortcomings were stated earlier in the manuscript, the reader expects a complete list in the Shortcomings paragraph. It is also not clear what is meant by, “the possibility of oversight of patients with mild symptoms”.

7. PLOS authors have the option to publish the peer review history of their article (what does this mean?). If published, this will include your full peer review and any attached files.

Reviewer #1: **Yes: **Richard Moon

---

## [Editor Report · Acceptance letter]

11 Jun 2021

PONE-D-20-36680R2 

Use of hyperbaric oxygen therapy for preventing delayed neurological sequelae in patients with carbon monoxide poisoning: A multicenter, prospective, observational study in Japan 

Dear Dr. Fujita:

I'm pleased to inform you that your manuscript has been deemed suitable for publication in PLOS ONE. Congratulations! Your manuscript is now with our production department. 

Kind regards, 

on behalf of

Dr. Tai-Heng Chen 

Academic Editor

PLOS ONE